# COVID-19 Vaccine Confidence Mediates the Relationship between Health Literacy and Vaccination in a Diverse Sample of Urban Adults

**DOI:** 10.3390/vaccines11121848

**Published:** 2023-12-13

**Authors:** Emily Hurstak, Francesca R. Farina, Michael K. Paasche-Orlow, Elizabeth A. Hahn, Lori E. Henault, Patricia Moreno, Claire Weaver, Melissa Marquez, Eloisa Serrano, Jessica Thomas, James W. Griffith

**Affiliations:** 1Section of General Internal Medicine, Boston University Chobanian & Avedisian School of Medicine, Boston, MA 02119, USA; lori.henault@bmc.org; 2Department of Medical Social Sciences, Northwestern University Feinberg School of Medicine, Chicago, IL 60611, USA; francesca.farina@northwestern.edu (F.R.F.); e-hahn@northwestern.edu (E.A.H.); claire.weaver@northwestern.edu (C.W.); melissa.marquez@northwestern.edu (M.M.); eloisa.serrano@northwestern.edu (E.S.); jessica.thomas@northwestern.edu (J.T.); j-griffith@northwestern.edu (J.W.G.); 3Department of Medicine, Tufts University School of Medicine, Tufts Medical Center, Boston, MA 02116, USA; mpo@tufts.edu; 4Department of Public Health Sciences, University of Miami Miller School of Medicine, Miami, FL 33136, USA; patricia.moreno@miami.edu

**Keywords:** vaccine confidence, health literacy, vaccine hesitancy, COVID-19

## Abstract

We sought to analyze the relationship between health literacy, confidence in COVID-19 vaccines, and self-reported vaccination. We hypothesized that the relationship between health literacy and vaccination would be mediated by vaccine confidence. We recruited (N = 271) English- and Spanish-speaking adults in Boston and Chicago from September 2018 to September 2021. We performed a probit mediation analysis to determine if confidence in COVID-19 vaccines and health literacy predicted self-reported vaccination. We hypothesized that the relationship between health literacy and vaccination would be mediated by vaccine confidence. Participants were on average 50 years old, 65% female, 40% non-Hispanic Black, 25% Hispanic, and 30% non-Hispanic White; 231 (85%) reported at least one COVID-19 vaccination. A higher mean vaccine confidence score (*t* = −7.9, *p* < 0.001) and higher health literacy (*t* = −2.2, *p* = 0.03) were associated with vaccination, but only vaccine confidence predicted vaccination in a multivariate model. Vaccine confidence mediated the relationship between health literacy and COVID-19 vaccination (mediated effects: 0.04; 95% CI [0.02, 0.08]). We found that using a simple tool to measure vaccine confidence identified people who declined or delayed COVID-19 vaccination in a diverse sample of adults with varying levels of health literacy. Simple short survey tools can be useful to identify people who may benefit from vaccine promotion efforts and evidence-based communication strategies.

## 1. Introduction

Despite the extensive evidence for COVID-19 vaccine effectiveness [1,2], vaccine promotion efforts [3], and broad vaccine availability, vaccination rates continue to be below optimal [4,5]. Concurrently, there are disparities in COVID-19 vaccination rates across different sociodemographic groups [6,7,8] and rural compared to urban regions [9,10,11]. These trends have persisted and will influence future pandemics [12,13].

Many factors influence vaccination acceptance, including confidence in efficacy and safety, the perception of risks and benefits, and availability, as well as political, cultural, and religious factors [14,15,16]. Vaccine confidence is defined as the degree of trust individuals have in a specific vaccine and the health care system recommending vaccination [17]. Not surprisingly, lower vaccine confidence is associated with a lower likelihood of vaccination [18,19]. Confidence also influences vaccine hesitancy [20], a behavioral state inclusive of both complete vaccine refusal and delays in vaccination [14]. Given the potential of alienating individuals with uncertainly around vaccination, shifting public health focus away from hesitancy and towards confidence may be less stigmatizing to individuals with vaccination concerns [21,22]. Additionally, interventions to increase vaccine confidence can emphasize the role of individual decision making through empathetic patient-centered discussions of risks and benefits, rather than focusing on reducing a less desirable state (i.e., hesitancy) [23].

Various tools have been used to measure vaccine confidence. The Vaccine Confidence Index (VCI; www.vaccineconfidence.org) measures individual perceptions of vaccine efficacy and safety [24] and has been adapted to different settings and infectious diseases [25,26,27]. Given the factors associated with lower vaccine uptake vary by community and region [28], measuring vaccine confidence in different settings can help to identify communities that may benefit from vaccine education interventions and outreach. In addition, understanding measures of vaccine confidence that perform well in diverse samples will also inform surveillance efforts [29]. Vaccination experts have called for improving the scientific understanding of the determinants of confidence and trust in vaccines, emphasizing that scientific inquiry into these factors is as important as studies that evaluate the efficacy and safety of vaccines themselves [30]. Health literacy has been defined as an individual’s ability to find, understand, and use information to inform their health decisions [31]. Health literacy has previously been shown to be related to vaccination attitudes [32,33]. We have previously shown that low health literacy was associated with low vaccine confidence. Specifically, health literacy mediated the relationship between race and ethnicity, and vaccine confidence [34]. Here, we assessed the relationship between health literacy, vaccine confidence, and vaccination. We hypothesized that higher health literacy and vaccine confidence would be associated with higher vaccination, and that confidence would mediate the relationship between health literacy and vaccination.

## 2. Materials and Methods

Our methods have been previously described [34].

### 2.1. Setting, Recruitment, and Population

Data for this analysis were collected within a parent study that sought to evaluate if health literacy impacts the psychometric properties of frequently used health questionnaires (clinical trial number: NCT03584490). Potential participants were adults in Boston and Chicago, recruited through community-based outreach and promotion on Research Match (www.researchmatch.org). Eligibility included an age of 18+ years; Spanish or English speaking; and ability to provide informed consent. Following informed consent procedures, participants were interviewed at baseline, 3 months, and 6 months by bilingual research staff. Participants were remunerated for participation.

At a baseline in-person visit (September 2018–March 2020), participants were characterized in terms of demographics, health literacy, and other screening measures. After the onset of the COVID-19 pandemic, visits were conducted via phone (April 2021–September 2021). Participants completed multiple health questionnaires during phone visits of a 60–90 min duration. The sample was restricted to English and Spanish speakers as these are the two most common languages spoken in the geographic regions of the study and given that the study assessment tools were validated in these two languages. Individuals with impairments in cognition, vision, or hearing that would prevent the completion of survey questionnaires were excluded from participation. Procedures were approved by Northwestern and Boston University Medical Center’s Institutional Review Boards.

### 2.2. Variables

At baseline, self-reported demographic data including age, gender, race and ethnicity, preferred language, and education level were collected. We assessed health literacy using Health Literacy Assessment Using Talking Touchscreen Technology (Health LiTT), which is a self-administered, computerized, performance-based measure that has been validated in English and Spanish and is scored on a T-score scale [35,36,37]. Higher scores indicate higher levels of overall health literacy. Health LiTT assesses performance-based health literacy skills including prose, document, and quantitative literacy.

Vaccine confidence was assessed at baseline, 3 months, and 6 months using an adapted, eight-item VCI (aVCI) [25]. The adapted VCI was inspired by the Global Vaccine Confidence Index, a survey-based measure of confidence in immunizations used extensively in global samples [15,24,38]. The adapted VCI is a more focused survey including eight Likert-style questions related to vaccination views (aVCI text, Appendix B). Respondents chose their level of agreement with responses (totally agree to totally disagree) with a score range of 0.25–4.0. Higher scores on the aVCI indicate a higher ratio of vaccine confidence to vaccine-related concerns. We measured vaccination status at 6 months using one item, “Have you received at least one dose of a vaccine for COVID-19?” with a Yes/No response. Vaccinations became available in the study areas in April 2021 [39,40,41].

### 2.3. Analysis

Data analyses were completed in R (v4.2.2) and Mplus (v8.9). For the analyses, race and ethnicity were categorized into 4 groups as White Non-Hispanic and other (including participants designating themselves as more than one race), Asian Non-Hispanic, Black Non-Hispanic, or Hispanic/LatinX. Education was categorized as (1) 12th grade-equivalent or less, (2) some college level education (incomplete) or an associate or technical degree, or (3) a college degree or higher level of education.

We compared aVCI and health literacy scores for participants who were vaccinated and those who were not using two sample *t*-tests. A probit mediation analysis in Mplus [42] was used to test if baseline vaccine confidence mediated the relationship between health literacy and vaccination. Maximum likelihood estimation with bootstrapping (5000 iterations) and Monte Carlo integration were used for the mediation analysis. We hypothesized that health literacy would predict higher odds of vaccination, and that this relationship would be mediated by vaccine confidence.

To account for the possibility of non-random missing data for vaccine confidence (i.e., where skipped items indicate lower vaccine confidence), the analysis was repeated with ‘missingness’ as a covariate, defined as Yes/No for participants who skipped any aVCI items. We hypothesized that skipping items would predict a lower probability of vaccination. To examine potential changes in confidence over time, we conducted a latent growth curve analysis with health literacy measured at baseline and vaccine confidence at baseline, 3 months, and 6 months. We hypothesized that increases in the intercept and slope of vaccine confidence would predict vaccination. The results of the missingness analysis are included in our Appendix A.

## 3. Results

### 3.1. Sample

Of the total sample (*N* = 302), 31 participants did not provide vaccination status and were excluded, resulting in an analytic sample of 271 participants (Table 1). The sample was predominantly female (65.0%) and non-Hispanic (74.8%) with a mean age of 50.2 years (±16.2). No participants identified as transgender or non-binary. Most participants identified as either Black (39.5%) or White (29.5%) and had at least some college education (61.6%). Mean (SD) health literacy was 53.5 ± 8.5 (T score units; a score of 55 is the cutoff for adequate health literacy). The mean (SD) baseline aVCI ratio score was 2.4 ± 1.0. Most participants (85.2%) reported being vaccinated. Participants who reported vaccination had higher aVCI (*p* < 0.001, *d* = 1.3) and higher health literacy (*p* = 0.030, *d* = 0.4; Figure 1) scores. There were statistically significant differences in vaccination for Black participants compared to White participants (not vaccinated: 62.5% Black vs. 12.5% White; *p* < 0.001; *d* = 0.5).

### 3.2. Vaccination Prediction

In univariate analyses, both vaccine confidence and health literacy predicted vaccine acceptance. The association between vaccine confidence and vaccination was positive, such that higher confidence scores were associated with higher odds of vaccination. Health literacy was not a significant predictor of vaccination, after controlling for vaccine confidence (Table 2). In path analysis, vaccine confidence mediated the relationship between health literacy and vaccination (Table 2, Figure 2). In the latent growth curve analysis, a change in vaccine confidence was not a significant predictor of vaccination (Table 3). Missing any responses on the aVCI also did not predict vaccination (results included in Appendix A).

## 4. Discussion

In a sample of diverse adults in Boston and Chicago, COVID-19 vaccination differed according to race, ethnicity, and health literacy. Vaccine confidence predicted vaccination and completely mediated the association between health literacy and vaccination. These results combined with those of an earlier study [34] demonstrate that health literacy is an important driver of vaccine confidence, but that a measure of confidence is a more specific predictor of vaccination. This result is consistent with findings from other studies demonstrating that vaccine-specific literacy and perceived health status may be the most important predictors of vaccination [43,44]. Thus, using simple measures of vaccine literacy or vaccine confidence can provide actionable information to tailor interventions towards increasing vaccination among populations at risk for disparities, including individuals with varying health literacy levels.

Strategic assessments of vaccine confidence can help public health entities anticipate evolving infectious disease threats and address the infodemic of misinformation that contributes to low vaccine confidence [45]. Vaccine confidence has been shown to evolve over time even among people who have previously accepted vaccinations [46,47]. Although we did not identify a relationship between vaccine confidence over time and vaccination, the short duration of our follow up may have reduced our ability to detect differences.

Vaccine confidence may provide information about health behaviors beyond the likelihood of accepting a specific vaccination [48]. Vaccine confidence for one vaccine may predict low confidence for others [49]. In addition, low confidence has been associated with susceptibility to misinformation, lower trust in health care, and other negative health consequences [50,51]. Accurately tracking vaccine confidence in diverse communities will inform public health efforts to design more effective strategic and targeted interventions that improve vaccination uptake.

Efforts to increase vaccine confidence through education campaigns, health care provider training, and strategies to counteract misinformation are needed [52,53]. Collaboration with community partners like faith-based organizations have been shown to be effective [54]. Given that health literacy is an antecedent of vaccine confidence [34], interventions to improve health literacy may be an essential component of efforts that reduce susceptibility to misinformation [55,56] and increase vaccine confidence. Specifically, the development and evaluation of interventions that increase media health literacy are greatly needed due to the volume of conflicting information about the COVID-19 virus and vaccination [57,58,59].

Our study had limitations. The sample size calculations were designed around the primary outcome of the parent study, defined as the differential functioning of health questionnaires. The analysis presented here is a secondary analysis of an outcome measure added during the COVID-19 pandemic, and thus, results should be interpreted with caution. We assessed self-reported vaccination, increasing the possibility of social desirability bias. However, our analysis showing that aVCI missingness did not change our findings supports their robustness. Our sample recruited urban English and Spanish speakers and thus, the results may not be generalizable to other populations. Data collection in additional populations is needed. The health literacy measure used in this study, Health LiTT, is a test-based measure of functional health literacy; it does not measure other dimensions of health literacy relating to constructs such as information seeking and behavioral interaction. Potentially, dimensions of health literacy not examined in this analysis would be stronger predictors of vaccine acceptance. For example, media literacy is likely to have significant relevance during the COVID-19 pandemic given the abundance of online information about COVID-19 vaccines and treatments [60]. Finally, some have advocated for measures of vaccine confidence to assess health care trust at the system and provider levels [16,27]. The aVCI tool we used was brief, specific to COVID-19, and did not provide such an assessment. Incorporating measures of trust should be considered in future research.

## 5. Conclusions

The analysis described here identified significant differences in COVID-19 vaccine acceptance by race, ethnicity, and health literacy. However, the association between health literacy and COVID-19 vaccine acceptance is entirely mediated by vaccine confidence scores. Thus, we found that using a simple tool to measure vaccine confidence identified people who declined or delayed COVID-19 vaccination in a diverse sample of adults with varying levels of health literacy. This implies that simple short survey tools can be useful to identify people who may benefit from directed vaccine promotion efforts and evidence-based communication strategies. Assessments of vaccine confidence can be conducted to improve public health surveillance and facilitate strategies that increase confidence and vaccination, particularly for communities affected by health disparities.

## Figures and Tables

**Figure 1 vaccines-11-01848-f001:**
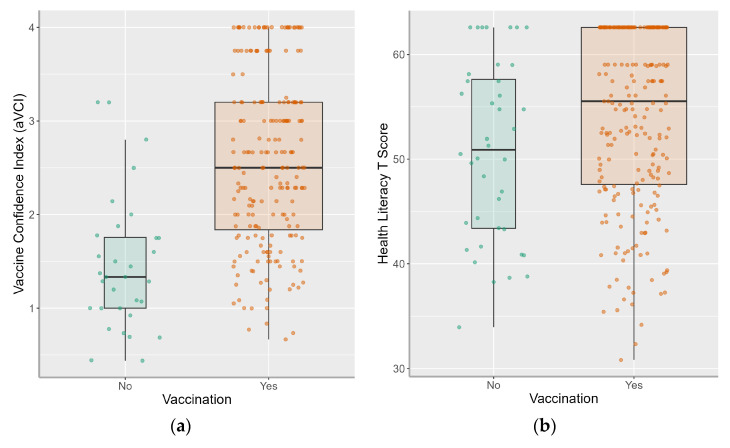
Boxplots of vaccination status by vaccine confidence (**a**) and health literacy (**b**). The box’s centerline is the median; upper and lower quartile scores bound the box. Line extension “whiskers” extend from minimum to maximum scores.

**Figure 2 vaccines-11-01848-f002:**
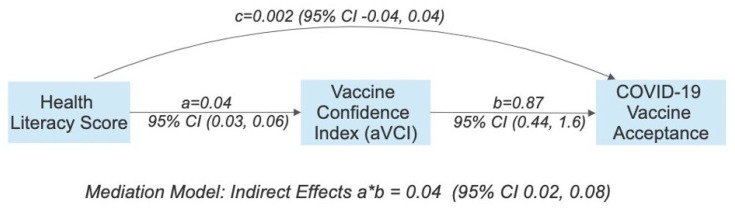
Mediation model for health literacy, vaccine confidence, and COVID-19 vaccination. Note: values are rounded to two decimal places for visualization purposes.

**Table 1 vaccines-11-01848-t001:** Participant demographic characteristic at baseline.

	Vaccinated (n = 231)	Not Vaccinated (n = 40)	Total (n = 271)
**Age (mean ± SD)**	50.5 (±16.6)	48.3 (±13.4)	50.2 (±16.2)
**Gender (female, %)**	149 (64.5%)	27 (67.5%)	176 (65.0%)
**Race and ethnicity (n, %)**			
**Asian**	10 (4.3%)	0	10 (3.7%)
**Black**	82 (35.5%)	25 (62.5%)	107 (39.5%)
**Hispanic**	59 (25.5%)	9 (23.1%)	68 (25.2%)
**Mixed or other**	5 (2.2%)	1 (2.5%)	6 (2.2%)
**White**	75 (32.5%)	5 (12.5%)	80 (29.5%)
**Education (n, %)**			
**≤12th grade**	86 (37.2%)	18 (45.0%)	104 (38.4%)
**Some college**	44 (19.1%)	11 (20.3%)	55 (20.3%)
**College and above**	101 (43.7%)	11 (27.5%)	112 (41.3%)
**Health literacy** **(T Score mean ± SD)**	54.0 (±8.4)	50.8 (±8.5)	53.5 (±8.5)
**Vaccine confidence (aVCI)**			
**Baseline**	2.6 (±0.9)	1.5 (±0.7)	2.4 (±1.0)
**3 months**	2.9 (±0.9)	1.8 (±0.9)	2.8 (±1.0)
**6 months**	3.0 (±0.9)	1.8 (±0.9)	2.8 (±1.0)
**Missing aVCI items**	49 (21.2%)	11 (27.5%)	60 (22.1%)

No participants identified as transgender or non-binary. Some college education includes an incomplete bachelor’s degree or an associate degree, or a qualification from a technical school.

**Table 2 vaccines-11-01848-t002:** Mediation model estimates including direct and indirect effects.

	Path	Estimate	95% Bootstrapped Confidence Intervals
**Health literacy**	**Direct effects**		
(continuous)	Health literacy ==> Vaccination	0.00	−0.04, 0.04
**Vaccine confidence**	**Mediator ==> Vaccination**		
(aVCI, continuous)	aVCI ==> Vaccination	0.87 *	0.44, 1.58
	**Mediator ==> Health literacy**		
	aVCI ==> Health literacy	0.04 *	0.03, 0.06
	**Indirect effects on Vaccination**		
	Health literacy ==> aVCI ==> Vaccination	0.04 *	0.02, 0.08

aVCI = Adapted Vaccine Confidence Index; Estimates are unstandardized. Estimates are starred (*) if the 95% confidence interval does not contain zero; Vaccine confidence is the mediator.

**Table 3 vaccines-11-01848-t003:** Prediction of Vaccination by Health Literacy and Vaccine Confidence (baseline and over time).

Predictor	Estimate (Unstandardized)	Estimate (Standardized)	*p*	95% CI
Health literacy	0.00	0.01	0.40	−0.02, 0.04
aVCI baseline	1.20	0.85	<0.001 ***	0.64, 1.05
aVCI change	−0.68	−0.09	0.80	−0.72, 0.54

Notes: aVCI = Adapted Vaccine Confidence Index; CI = confidence intervals; *** *p* < 0.001. Vaccination is the outcome.

## Data Availability

The authors have included the primary data used for this analysis, as well as Appendix A describing the analytic procedures and statistical code.

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
