# Peer review of "COVID-19 Vaccine Confidence Mediates the Relationship between Health Literacy and Vaccination in a Diverse Sample of Urban Adults"

_vaccines, 2023, doi:10.3390/vaccines11121848_

Round 1

Reviewer 1 Report

Comments and Suggestions for Authors

The paper is topical and interesting. The title reflects the content of the manuscript. The abstract can stand alone. The tables and figures are clear and appropriate.

In the manuscript, there are only two points that require the Authors' comments/revisions.:

Methods: despite the methods described in another study conducted by the Authors themselves, it should be important to provide more details here, to facilitate the reader, such as the modalities of administration of the questionnaire and collection of informed consent. and, regarding statistics. the definition of the sample size.

Discussion/limitations: the reviewer is not familiar with Litt, but from the literature description it seems that it is intended primarily for the assessment of functional health literacy. During the pandemic, it was observed by different Authors using other tools, that functional health literacy was often lower than interactive literacy, as if the Covid-related infodemic stimulated motivation and abilities to seek accurate information,  while the functional skills were challenged by the complexity and technicality of new medical definitions. Thus, it is possible that assessing interactive-critical literacy could be a better predictor of vaccine acceptance than functional. Authors may want to comment on this.

Author Response

Thank you for reviewing our manuscript. We appreciate your feedback. Please see the attached document with a point by point response to the feedback.

Reviewer 2 Report

Comments and Suggestions for Authors

Dear authors,

It was a pleasure to read your manuscript. The manuscript gives another aspect of your parent study. I have two minor suggestions. The first is to include results regarding the relationship between vaccine confidence over time and vaccination in the main manuscript instead of supplementary material. The second is to give a more detailed explanation of aVCI.

Author Response

Thank you for reviewing our manuscript. We appreciate your feedback. Please see the attached document for a point by point response.

Reviewer 3 Report

Comments and Suggestions for Authors

In the present study, the authors describe the relationship between health literacy, vaccine confidence, and vaccination. The experimental design is simple, and the method has been used successfully to determine this link.

Title: I think the title well reflects the main aim and findings of the work.

I suggest to avoid the extensive use of personal forms (i.e., our, we etc.) throughout the study.

The abstract adequately summarizes the results and significance of the study, and the keywords represent the article adequately. However, the Authors should add some more information on the statistical analysis applied.

The introduction section is well written and it falls within the topic of the study, and Authors cited appropriately bibliographic information.

Line 48: The font size is not uniform.

Line 51: The font size is not uniform. Use a uniform font size throughout article.

The section of Materials and Methods is clear for the reader, and it meticulously describes the methods applied in the study. However, some information should be added.

Why did the authors choose only Spanish and English and not from other nations? Is there a particular reason for this?

Results section, as well as the discussion section, are clear and well-written, and the findings obtained in the study were well-discussed and justified with appropriate references.

In the conclusion section, the authors need to summarize the main results of the study and then emphasize the significance.

Tables and figures are generally good, representing well the results gathered in the study.

Comments on the Quality of English Language

A minor edit of the English language is required.

Author Response

Thank you very much for reviewing our manuscript. We feel we have greatly improved the text responding to your feedback. Please see the attached document for line by line responses. 

Reviewer 4 Report

Comments and Suggestions for Authors

Well done. No comments.

Author Response

Thank you for taking the time to review our manuscript and provide feedback.